# Rethinking Personalization in Large Language Models at the Token Level

## Abstract

With large language models (LLMs) now performing strongly across diverse tasks, there is growing demand for them to personalize outputs for individual users. Personalization is typically framed as an additional layer on top of a base NLP task, requiring model responses to meet user-specific needs while still accomplishing the underlying task. From a token-level perspective, different tokens in a response contribute to personalization to varying degrees. Tokens with higher personalization relevance should therefore receive greater emphasis when developing personalized LLMs. However, accurately estimating such personalization degrees remains challenging. To address this challenge, we propose PerContrast, a self-contrast method that estimates each output token's dependence on user-specific information through causal intervention. Building on this mechanism, we develop the PerCE loss, which adaptively up-weights tokens with higher estimated personalization degrees during training via a bootstrap procedure, enabling the model to alternate between estimating and optimizing these tokens. Experiments on multiple LLMs demonstrate that PerCE substantially improves personalization performance with minimal additional cost, achieving average gains of over 10% and up to 68.04% on the LongLaMP dataset, along with strong cross-task and cross-scenario transferability. These results highlight the importance of token-level personalization modeling and establish token-aware training as a simple yet effective paradigm for advancing personalized LLMs.

[1]Anonymous Institution, Anonymous City, Anonymous Region, Anonymous Country. Correspondence to: Anonymous Author <anon.email@domain.com>.

Preliminary work. Under review by the International Conference on Machine Learning (ICML). Do not distribute.

## 1. Introduction

Large language models (LLMs) have demonstrated remarkable capabilities across a wide range of tasks, such as dialogue (Bubeck et al., 2023; Ding et al., 2023), question answering (Trivedi et al., 2022; Salemi & Zamani, 2025), and logic reasoning (Guo et al., 2025; Trung et al., 2024). This progress has led to widespread deployment of LLMs in user-facing applications. As a result, there is a growing demand for LLMs to not only excel at general tasks, but also to *personalize* their outputs to individual users. In other words, beyond producing generally correct answers, users increasingly expect LLMs to tailor responses according to the user's profile, preferences, or interaction histories (Zhang et al., 2024; Xu et al., 2025b).

Many works have been devoted to this goal. One line of research focuses on synthesizing personalized data to train LLMs across diverse scenarios, including personalized text generation (Salemi et al., 2024b; Kumar et al., 2024), personalized question answering (Salemi & Zamani, 2025), and personalized dialogue (Wu et al., 2024). Another line of research seeks to enhance personalization by developing more effective methods for retrieving, modeling, and integrating user-specific information into LLMs (Salemi & Zamani, 2025; Salemi et al., 2025b; Liu et al., 2025). Collectively, these efforts have substantially advanced personalized LLMs. However, a fundamental characteristic of personalization tasks has so far been largely overlooked.

Unlike other tasks, personalization is typically framed as an additional layer on top of a base NLP task, requiring model responses to address user-specific needs while still accomplishing the underlying task (Salemi et al., 2024b; Kumar et al., 2024; Zhao et al., 2025; Xu et al., 2025a). From a token-level perspective, different tokens in a response contribute to personalization to varying degrees, as shown in Figure 1. Naturally, these tokens should receive different emphasis when developing personalized LLMs. However, existing training practices typically treat all tokens uniformly (Kumar et al., 2024; Salemi & Zamani, 2025; Tan et al., 2024), which may dilute the emphasis on personalization.

A key challenge is how to obtain token-level personalization degrees. The distribution of personalization importance across tokens is difficult to characterize, as different types

*Figure 1.* Tokens contribute to personalization to varying degrees, and their contribution distributions differ across tasks. In text generation tasks, stylistic tokens play a more prominent role, whereas in conversational tasks, information-bearing tokens are more important.

of tasks impose distinct requirements on personalization. As illustrated in Figure 1, in personalized abstract generation or topic writing, personalization is primarily reflected through stylistic tokens, whereas in personalized dialogue, it is more closely associated with tokens encoding individual traits. This task-dependent variability further complicates the estimation of token-level personalization importance.

To overcome this limitation, we introduce **PerContrast**, a principled method for personalization degree estimation. The core idea is to measure how much each token in the task response depends on user-specific information in the prompt via causal intervention. Concretely, for a given response token, we compare the model's predicted probability of that token when conditioned on the full personalized instruction versus a modified instruction with the personal information removed. We further prove that this likelihood difference corresponds to the token-level causal effect. This causal analysis enables PerContrast to quantify the degree to which each token contributes to personalization.

We believe that the mechanism for identifying personal tokens can greatly benefit training for personalization. To this end, we propose the **PerCE** loss, which upweights important tokens that are estimated by the model itself. In this way, PerCE bootstraps personalization by alternating between estimating and optimizing personal tokens. This training mechanism is orthogonal to existing personalized LLM training pipelines (Salemi et al., 2024b; Kumar et al., 2024), and can be integrated with them to further enhance personalization. Experimental results across multiple LLMs show that PerCE consistently outperforms the conventional Cross Entropy (CE) loss with minimal additional cost. It achieves a maximum gain of 68.04% and an average improvement of over 10% across all models on LongLaMP dataset. Moreover, PerCE demonstrates significantly stronger transferability in personalization, achieving gains of up to 50% in many cross-task and cross-scenario settings.

In conclusion, the main contributions of this paper are:

- We conduct the first token-level analysis of personalization and introduce PerContrast, an efficient self-contrast method that quantifies each token's contribution to personalization with causality theoretical guarantee.

- We develop the PerCE loss, which enhances the model's personalization capabilities in an expectation–maximization (EM) manner. PerCE allows the model to alternately perform online self-contrast weighting and optimization at each training step, thereby encouraging it to automatically place greater focus on personalized information.

- We conduct extensive experiments across multiple models with scale from 4B to 14B on a wide range of settings. The results show that PerCE substantially improves the performance and generalization in personalization scenarios while incurring minimal additional cost.

## 2. PerContrast: Causal Measurement of Token-Level Personalization

In this section, we introduce **PerContrast**, a self-contrast method that quantifies each token's contribution to personalization. We first describe the core design of PerContrast and its causal motivation. We then exhibit that the proposed intervention in the PerContrast method leads to the token-level causal effect To evaluate its estimation accuracy, we conduct a synthetic experiment with controlled personalization signals. All experimental details and results are reported in Appendix A.

### 2.1. Measuring Tokens via Self-Contrast

LLM personalization aims to adapt generic models to user-specific preferences while maintaining strong performance on the base task. Unlike standard NLP tasks, where outputs

2

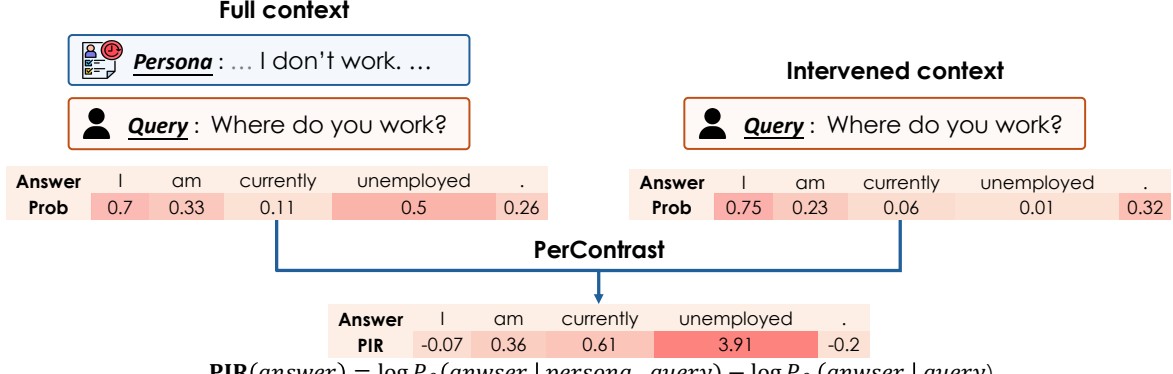

$$\mathbf{PIR}(answer) = \log P_\theta(anwser \mid persona, \ query) - \log P_\theta(anwser \mid query)$$

*Figure 2.* Illustration of PerContrast. By intervening on the user persona, PerContrast estimates the personalization degree of each output token. PIR denotes the Personal Influence Ratio (defined in Equation 2).

are conditioned only on the input query, personalization additionally depends on the user persona $p_u$. Formally, given an input query $x$ and a user persona $p_u$, a personalized LLM models the conditional probability distribution:

$$P_\theta(y \mid p_u, x), \tag{1}$$

where $y$ is the generated response and $\theta$ are the model parameters. For evaluation, the response is either compared with a reference response $y_r$ or assessed directly by an LLM judge.

As illustrated in Figure 1, different tokens contribute to personalization to varying degrees, and therefore warrant different levels of emphasis during training. However, existing research lacks a principled metric for quantifying personalization at the token level. To address this gap, we propose **PerContrast**, a principled method that measures the degree to which each token depends on user-specific information. This token-level personalization signal provides a foundation for assigning differentiated training emphasis, enabling more effective optimization for personalized LLMs.

**Personalization Intervention.** To quantify the influence of a user persona on each response token $y_i$, we perform an *intervention* on the model context. Specifically, given a personalization task with query–response pair $(x, y)$ and a language model $P_\theta$ (with strong personalization ability), we compute the difference between the log probability of $y_i$ with the full persona $p_u$ and the log probability without the persona:

$$\mathrm{PIR}(y_i; \theta) = \log P_\theta(y_i \mid \textcolor{orange}{p_u}, x, y_{<i}) - \log P_\theta(y_i \mid x, y_{<i}). \tag{2}$$

We refer to this measure as the **Personal Influence Ratio (PIR)**, which captures the contribution of the user persona to the prediction of each token.

From a causal perspective, the second term in Equation 2 serves as the counterfactual outcome obtained by the intervention (removing the user persona). Accordingly, PIR estimates the token-level causal effect (Pearl, 2010) of the persona under the language model $P_\theta$. A high PIR value indicates that the persona plays an important role in predicting $y_i$, thereby identifying it as a key token in personalization tasks, as shown in Figure 2.

### 2.2. A Causal-Theoretic Analysis of PIR

In this section, we formally analyze the PIR from the causal perspective. Specifically, we treat each personalization instance as one causal unit with a query $x$, a persona segment $p$, and a reference response $\mathbf{y} = (y_1, \ldots, y_n)$. For token position $i$, we focus on the prediction of the reference token $y_i$ given the prefix $y_{<i}$, $x_i$, and $p_i$. Figure 3 depicts the causal structure underlying personalization as a Directed Acyclic Graph (DAG). The rationale behind the DAG is that persona information $p$ is usually retrieved based on the query $q$, and both persona and query can affect the token-level response. In addition, to reflect the autoregressive nature of generation, we also specify the response prefix tokens $y_{<i}$, which influence the next-token generation.

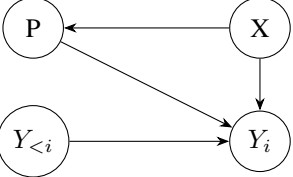

*Figure 3.* Causal Graph When Predict $Y_i$

Formally, under potential outcome framework (Rubin, 2005), for unit $i$, we define the $P_i$ as treatment, which has two values $p_i^M$ (embedding under Mask) and $p_i^{NM}$ (embedding under Non-Mask). We define the potential outcome $Y_i(p_i^{NM})$ / $Y_i(p_i^M)$ as the counterfactual next token gen-

3

erated under non-mask / mask scenario. We focus on the conditional probability that potential outcome equals $y_i$ with a language model $P_\theta$:

$$\mathbb{P}_\theta \left( Y_i(p_i) = y_i \mid X_i = x_i, Y_{<i} = y_{<i} \right).$$

Based on the above framework, we make the following standardized causal assumptions, including the no interference assumption and the unconfoundedness assumption.

**Assumption 2.1** (No Interference). The potential outcomes for one unit do not depend on interventions on other units.

The no interference assumption holds due to the next token prediction independents with other units.

**Assumption 2.2** (Unconfoundedness). For each unit $i$, we have $(Y_i(P_i^M), Y_i(P_i^{NM})) \perp P_i \mid X_i, Y_{<i}$.

The unconfoundedness assumption means that there is no unmeasured variable affect the treatment $P_i$ and outcome $Y_i$, which naturally holds because we condition on $X_i$. According to the above 2 assumptions, as well as the widely-used assumption in causal inference (Rubin, 2005), such as positivity, consistency, etc., we have

$$\mathbb{P}_\theta \left( Y_i(P_i = p_i) = y_i \mid X_i = x_i, Y_{<i} = y_{<i} \right)$$
$$= \mathbb{P}_\theta \left( Y_i(P_i = p_i) = y_i \mid X_i = x_i, Y_{<i} = y_{<i}, P_i = p_i \right)$$
$$= \mathbb{P}_\theta \left( Y_i = y_i \mid X_i = x_i, Y_{<i} = y_{<i}, P_i = p_i \right).$$

We focus on the token-level causal effect (CE), defined as the following log difference:

$$\mathrm{CE}(y_i; \theta) = \log \mathbb{P}_\theta \left( Y_i(p_i^M) = y_i \mid x_i, y_{<i} \right)$$
$$- \log \mathbb{P}_\theta \left( Y_i(p_i^{NM}) = y_i \mid x_i, y_{<i} \right)$$

Following the identification theoretical results of causal effect (VanderWeele, 2015; Pearl, 2022), we come to the below conclusion.

**Theorem 2.3** (Relation Between Causal Effect and PIR). *Under Assumptions 2.1-2.2, the token-level causal effect is the same as the proposed PIR*

$$\mathrm{CE}(y_i; \theta) = \mathrm{PIR}(y_i; \theta) \tag{3}$$

## 3. PerCE: Enhancing Personalized LLMs with an EM Perspective

Cross-Entropy (CE) remains the standard training objective for personalized LLMs (Salemi et al., 2024b; Tan et al., 2024). Given an input query $x$, a user persona $p_u$, and the reference response $y_r$, CE computes a uniform average over all tokens:

$$\mathrm{CE}(y; \theta) = -\frac{1}{n} \sum_{i=1}^{n} \log P_\theta(y_i \mid x, y_{<i}). \tag{4}$$

While this objective encourages the model to follow task instructions and incorporate user preferences, it implicitly assumes that all tokens contribute equally. This assumption is misaligned with the nature of personalization, where different tokens contribute to personalization to varying degrees and therefore warrant different levels of emphasis during training. Under the standard CE objective, this uniform averaging can obscure the influence of a small number of highly personalization-relevant tokens, ultimately limiting personalization performance.

**Weighted CE as a More Suitable Objective.** To address this issue, we introduce a set of token-level importance weights $w = (w(y_1), \ldots, w(y_n))$, representing how strongly each token contributes to personalization. A more appropriate objective is the weighted CE:

$$\mathrm{WCE}(y; \theta) = -\frac{1}{n} \sum_{i=1}^{n} w(y_i) \log P_\theta(y_i \mid p_u, x, y_{<i}). \tag{5}$$

However, the personalization importance $w(y_i)$ is not directly observable. We therefore need an estimator of these latent token weights.

**PIR as an Empirical Estimator of Token Importance.** The PIR metric (Equation 2) quantifies how strongly the user persona influences the model's predicted likelihood of token $y_i$. Tokens whose likelihood changes substantially when the persona is removed are more closely tied to personalization and therefore should receive larger importance weights. This makes PIR a natural empirical estimator for the latent token-level personalization importance:

$$\hat{w}(y_i; \theta) = \mathrm{clip}(\mathrm{PIR}(y_i; \theta), m, M), \tag{6}$$

where $[m, M]$ stabilizes training by avoiding extreme gradients.

**Causal Interpretation for token reweighting.** The causal analysis of PIR above provides a principled lens for understanding token reweighting. Within this framework, tokens with large causal effects naturally need more emphasis during training. PerCE adopts this idea by optimizing a weighted cross-entropy loss.

**An EM Interpretation of PerCE.** We now introduce **PerCE**, an online token-reweighted training objective designed to enhance personalization. The key idea is to treat personalization as a latent-variable problem, where the latent variable corresponds to the personalization importance of each token. PerCE iteratively estimates these token-level importances and then optimizes the model with respect to them, forming a procedure closely analogous to the Expectation–Maximization (EM) algorithm.

4

In the ***E-step (Estimating token importance)***, given the current model parameters $\theta^{(t)}$, we compute the PIR score for each token and convert it into a clipped importance weight:

$$w^{(t)}(y_i) \leftarrow \text{clip}\big(\text{PIR}(y_i; \theta^{(t)}), m, M\big). \qquad (7)$$

In the ***M-step (Optimizing the weighted objective)***, using the estimated weights, we update the model parameters by minimizing the weighted Cross-Entropy loss:

$$\theta^{(t+1)} = \arg\min_{\theta} -\frac{1}{n} \sum_{i=1}^{n} w^{(t)}(y_i) \log P_\theta(y_i \mid p_u, x, y_{<i}). \qquad (8)$$

Putting these steps together, PerCE forms an online EM loop: (i) the model infers token-level personalization influence via PIR (E-step), and (ii) the model updates itself using the weighted CE loss (M-step). This bootstrap mechanism progressively strengthens the model's ability to detect and amplify personal tokens without requiring any additional supervision or annotation.

## 4. Experiments

To evaluate PerCE, we compare its performance with three CE loss variants on the LongLaMP dataset across multiple LLMs. The results show that PerCE achieves significant improvements, with a maximum gain of 68.04% and an average improvement of over 10% across all models compared standard CE loss. Moreover, PerCE demonstrates strong generalization, achieving significant gains in many cross-task and cross-scenario settings. Importantly, these substantial gains come at minimal cost, since PerCE introduces only a slight overhead by requiring just one additional forward pass with a short persona-removed context. Additional results reported in Appendix C further corroborate the effectiveness and robustness of our method.

**Baseline.** PerCE is the first loss function specifically designed for personalized LLMs. For a comprehensive comparison, in addition to the standard CE loss, we include two additional baselines that have shown effectiveness in other domains such as reasoning and pretraining. LossCE (Lin et al., 2024a) assigns higher weights to tokens with larger prediction errors during training, while EntCE (Wang et al., 2025) upweights tokens with higher predictive entropy.

**Dataset.** To assess the effectiveness of our approach, we primarily use the LongLaMP dataset (Kumar et al., 2024), which is designed for personalized text generation. LongLaMP includes three tasks: Personalized Abstract Generation (PAG), Personalized Review Writing (PRW),

and Personalized Topic Writing (PTW).[1] Given the large scale of the dataset, we randomly sample 2,000 training examples and 500 test examples for each task. To evaluate cross-scenario transferability, we additionally conduct experiments on the ALOE benchmark (Wu et al., 2024), which consists of 100 carefully curated multi-turn conversations associated with distinct user personas. Unlike LongLaMP, where user persona information is explicitly provided in the prompt, ALOE does not directly supply user information, requiring models to infer user preferences implicitly through dialogue history. To further demonstrate the generality of our method, we also evaluate PerCE on several short-text personalization tasks from the LaMP benchmark (Salemi et al., 2024b).

**Metric.** For both datasets, we adopt the official evaluation metrics. For LongLaMP, we follow the official protocol and report ROUGE-L and METEOR to measure the quality of personalized text generation. For ALOE, evaluation is conducted with given LLM-as-a-Judge framework, where an LLM evaluator assesses the degree of alignment to user-specific preferences in multi-turn dialogue on a 1–5 scale.

**Setup.** We conduct experiments on Qwen3-4B, Qwen3-14B (Yang et al., 2025) and Llama3.1-8B-Instruct (Llama3-8B) (Dubey et al., 2024). All models are trained with both standard CE and our PerCE on each task of the LongLaMP dataset. We follow the standard RAG setting used in LongLaMP, retrieving the top 4 relevant user-history entries with Contriever and appending them to the prompt. In addition to evaluating on the corresponding task, we test each model on other tasks within LongLaMP to assess cross-task personalization transfer. Furthermore, we evaluate generalization in a distinct multi-turn personalized dialogue scenario to examine the model's cross-scenario personalization capability. Detailed hyperparameters and prompt templates are provided in Appendix B and D.

**Main Result.** As shown in Table 1, we report the performance of 3 models fine-tuned with 3 CE variants and PerCE on LongLaMP. The results show that PerCE yields an average improvement of over 10% compared with standard CE across all three tasks and all model scales. In particular, PerCE consistently outperforms all CE variants on Review Writing and Topic Writing, achieving METEOR score improvements of up to +68.04% and + 46.34% on Qwen3-4B. The performance on Abstract Generation are relatively less pronounced, likely because abstract generation is more constrained than open-ended writing tasks and thus offers fewer opportunities for personalization. Nevertheless, the consistently higher METEOR scores and subsequent hy-

---

[1]The dataset also contains a personalized email completion task, which relies on private data and is therefore excluded from our experiments.

*Table 1.* Performance (ROUGE-L and METEOR) of Qwen3-4B, Qwen3-14B and Llama3-8B fine-tuned with three weighted CE variants and PerCE on personalized generation tasks from LongLaMP, including Abstract Generation, Review Writing, and Topic Writing. Each model is trained separately on its corresponding task. The Gain row reports the improvement of PerCE over the standard CE.

| Model | Method | PAG | | PRW | | PTW | | Average | |
|---|---|---|---|---|---|---|---|---|---|
| | | ROUGE-L | METEOR | ROUGE-L | METEOR | ROUGE-L | METEOR | ROUGE-L | METEOR |
| Qwen3-4B | CE | **0.3727** | 0.3225 | 0.1898 | 0.1583 | 0.1665 | 0.1379 | 0.2430 | 0.2062 |
| | LossCE | 0.3508 | 0.2941 | 0.2183 | 0.1958 | 0.1783 | 0.1747 | 0.2491 | 0.2215 |
| | EntCE | 0.3564 | 0.2958 | 0.2193 | 0.1938 | 0.1761 | 0.1704 | 0.2506 | 0.2200 |
| | **PerCE** | 0.3619 | **0.3342** | **0.2668** | **0.2660** | **0.2102** | **0.2018** | **0.2796** | **0.2673** |
| | **Gain** | -2.90% | +3.63% | +40.57% | +68.04% | +26.25% | +46.34% | +15.06% | +29.63% |
| Qwen3-14B | CE | **0.3865** | 0.3389 | 0.2321 | 0.2556 | 0.1805 | 0.1522 | 0.2664 | 0.2489 |
| | LossCE | 0.3811 | 0.3309 | 0.2628 | 0.2348 | 0.1945 | 0.1608 | 0.2795 | 0.2422 |
| | EntCE | 0.3892 | 0.3355 | 0.2568 | 0.2279 | 0.1950 | 0.1619 | 0.2803 | 0.2418 |
| | **PerCE** | 0.3785 | **0.3479** | **0.2784** | **0.2717** | **0.2307** | **0.2207** | **0.2959** | **0.2801** |
| | **Gain** | -2.07% | +2.65% | +19.96% | +6.30% | +27.78% | +44.96% | +11.06% | +12.55% |
| Llama3-8B | CE | 0.3573 | 0.3238 | 0.2311 | 0.2143 | 0.2021 | 0.2081 | 0.2635 | 0.2487 |
| | LossCE | 0.3428 | 0.3217 | 0.2531 | 0.2156 | 0.2156 | 0.2216 | 0.2705 | 0.2530 |
| | EntCE | 0.3647 | 0.3329 | 0.2469 | 0.2347 | 0.2169 | 0.2043 | 0.2762 | 0.2573 |
| | **PerCE** | **0.3756** | **0.3511** | **0.2771** | **0.2713** | **0.2218** | **0.2316** | **0.2915** | **0.2847** |
| | **Gain** | +5.12% | +8.42% | +19.90% | +26.61% | +9.74% | +11.32% | +10.62% | +14.47% |

*Table 2.* Performance (ROUGE-L) across different learning rates for Qwen3-4B on the LongLaMP dataset. Green indicates the best results, while red indicates the worst. Variance values are scaled by $10^{-4}$. Personalized Abstract Generation (PAG), Personalized Review Writing (PRW), and Personalized Topic Writing (PTW) are abbreviated due to space limits.

| LR | Task | | |
|---|---|---|---|
| | PAG | PRW | PTW |
| 2e-6 | 0.3247 | 0.2446 | 0.1778 |
| 5e-6 | 0.3469 | 0.2590 | 0.1883 |
| 8e-6 | 0.3561 | 0.2619 | 0.1998 |
| 2e-5 | 0.3619 | 0.2650 | 0.2098 |
| 5e-5 | 0.3585 | 0.2668 | 0.2102 |
| Mean | 0.3496 | 0.2595 | 0.1972 |
| Variance | 1.800 | 0.6231 | 1.580 |

*(a)* PerCE

| LR | Task | | |
|---|---|---|---|
| | PAG | PRW | PTW |
| 2e-6 | 0.3560 | 0.1705 | 0.1417 |
| 5e-6 | 0.3728 | 0.1805 | 0.1542 |
| 8e-6 | 0.3687 | 0.1862 | 0.1379 |
| 2e-5 | 0.2339 | 0.1898 | 0.1665 |
| 5e-5 | 0.2261 | 0.1722 | 0.1532 |
| Mean | 0.3115 | 0.1798 | 0.1507 |
| Variance | **44.65** | 0.571 | 1.026 |

*(b)* CE

perparameter robustness analysis (Table 2) shows that by emphasizing personal tokens, PerCE also improves training stability. Overall, these results highlight that PerCE effectively enhances personalized text generation across different models and tasks.

**Robustness to Learning Rates.** As shown in Table 2, PerCE consistently delivers stable and strong performance across all tasks under varying learning rates. In contrast, while CE performs reasonably well on the PAG task, it is highly sensitive to learning rate changes: its ROUGE-L score drops sharply from 0.3728 to 0.2261 when the learning rate increases from $5e^{-6}$ to $5e^{-5}$, with variance reaching 44.65, indicating severe instability. These results demonstrate that PerCE not only achieves higher average performance but also provides substantially greater robustness to

hyperparameter variation.

**Cross-Task Generalization.** Table 3 and Table 8 present the cross-task transfer performance of Qwen3-4B and Llama3-8B fine-tuned with standard CE and PerCE, where models are trained on a single source task and evaluated on other target tasks. The results demonstrate that PerCE exhibits strong generalization: even when trained on one task, it consistently improves performance on other tasks compared to CE. For example, on Qwen3-4B trained on PTW, PerCE achieves gains of +56.62% and +49.90% on PAG and PRW over standard CE. Notably, several out-of-domain scores achieved by PerCE even surpass the in-domain scores of CE on the corresponding tasks, such as 0.2211 for PAG → PRW vs. 0.1898 for PRW, and 0.1955 for PRG → PTW vs. 0.1665 for PTW. These results highlight that PerCE

6

*Table 3.* Cross-task transfer performance (ROUGE-L) of Qwen3-4B fine-tuned with standard CE and PerCE. Rows indicate the source task used for training, while columns represent the target tasks used for evaluation. Detailed results for Llama3-8B are provided in Table 8.

| Source Task | Method | Target Task | | |
|---|---|---|---|---|
| | | PAG | PRW | PTW |
| PAG | CE | **0.3727** | 0.1744 | 0.1617 |
| | PerCE | 0.3619 | **0.2211** | **0.1657** |
| | **Gain** | -2.90% | +26.78% | +2.47% |
| PRW | CE | 0.2055 | 0.1898 | 0.1522 |
| | PerCE | **0.3290** | **0.2668** | **0.1955** |
| | **Gain** | +60.10% | +40.57% | +28.45% |
| PTW | CE | 0.2063 | 0.1443 | 0.1665 |
| | PerCE | **0.3231** | **0.2163** | **0.2102** |
| | **Gain** | +56.62% | +49.9% | +26.25% |

*Table 4.* Cross-scenario transfer performance of Qwen3-4B models fine-tuned with CE and PerCE. Rows correspond to the source task used for training, and columns indicate the performance of dialogue turns $k$ in the ALOE evaluation. Detailed results for Llama3-8B are provided in Table 9.

| Source Task | Method | ALOE | | | | | |
|---|---|---|---|---|---|---|---|
| | | k=6 | k=7 | k=8 | k=9 | k=10 | Average |
| PAG | CE | **2.55** | 2.52 | 2.35 | 2.33 | 2.40 | 2.43 |
| | PerCE | **2.55** | **2.55** | **2.60** | **2.52** | **2.60** | **2.56** |
| | **Gain** | 0.00 | +0.03 | +0.25 | +0.19 | +0.20 | +0.13 |
| PRW | CE | 1.25 | 1.23 | 1.32 | 1.27 | 1.27 | 1.27 |
| | PerCE | **2.83** | **2.80** | **2.65** | **2.75** | **2.85** | **2.78** |
| | **Gain** | +1.58 | +1.57 | +1.33 | +1.48 | +1.58 | +1.51 |
| PTW | CE | 1.07 | 1.05 | 1.05 | 1.07 | 1.12 | 1.07 |
| | PerCE | **2.80** | **2.90** | **2.70** | **2.67** | **2.73** | **2.76** |
| | **Gain** | +1.73 | +1.85 | +1.65 | +1.60 | +1.61 | +1.69 |

achieves robust and superior cross-task generalization.

**Cross-Scenario Transfer.** Personalization needs are inherently cross-scenario, as users expect models to generate preference-aligned outputs regardless of the context. Therefore, cross-scenario transfer is particularly important. To assess this ability, we evaluate the fine-tuned models on the ALOE benchmark. Unlike LongLaMP, where user history is explicitly included in the prompt, ALOE does not directly provide user information. Instead, it allows models to interact with users in multi-turn dialogue and requires them to infer preferences solely from the conversation in order to deliver user-tailored responses. This creates a substantial scenario gap between the two benchmarks. As shown in Table 4 and Table 9, PerCE consistently outperforms the standard CE loss across nearly all settings, achieving substantial gains and demonstrating significantly stronger cross-scenario personalization transfer. In addition, Table 11 shows that PerCE better preserves the model's general capabilities than standard CE, indicating that improvements in personalization do not compromise general language understanding.

standing.

**Training Efficiency.** Throughout the entire training process, PerCE significantly outperforms alternative loss functions. The only additional cost introduced by PerCE is a single extra forward pass per training step. Given the substantial improvements in both performance and generalization brought by PerCE, this overhead is entirely acceptable. Specifically, the extra forward pass in PerCE operates only on a short persona-removed context. In most personalization tasks, user persona information occupies a large portion of the context window. For LongLaMP, removing the persona reduces the input length by approximately 7%, as shown in Figure 6. Consequently, PerCE incurs minimal computational and time overhead in practice, making it well suited for real-world deployment.

**Results on LaMP.** To further demonstrate the generality of our method, we also conduct experiments on the short-text generation tasks in LaMP (Salemi et al., 2024b). It is intuitive that PerCE benefits long-text generation more, as

7

*Table 5.* Results of Qwen3-4B on three short-text personalization tasks from LaMP (Salemi et al., 2024b), including Personalized News Headline Generation (PNHG), Personalized Scholarly Title Generation (PSTG), and Personalized Tweet Paraphrasing (PTP).

| Method | PNHG | | PSTG | | PTP | | Average | |
|---|---|---|---|---|---|---|---|---|
| | ROUGE-L | METEOR | ROUGE-L | METEOR | ROUGE-L | METEOR | ROUGE-L | METEOR |
| CE | 15.3 | 16.5 | 25.6 | 42.0 | 34.6 | 44.6 | 25.17 | 34.37 |
| **PerCE** | **16.1** | **17.9** | **27.2** | **43.1** | **35.5** | **46.0** | **26.27** | **35.67** |
| **Gain** | +5.23% | +8.48% | +6.25% | +2.62% | +2.60% | +3.14% | +4.37% | +3.78% |

*Table 6.* Prompt length with and without user persona across three personalized tasks, tokenized by Qwen3Tokenizer.

| Setting | PAG | PRW | PTW | Average |
|---|---|---|---|---|
| w. Persona | 975.1 | 2062.6 | 1055.3 | 1364.3 |
| w.o. Persona | **64.5** | **168.9** | **54.8** | **96.1** |

longer responses contain more tokens and emphasis is more crucial during training. Nevertheless, as shown in Table 5, PerCE consistently outperforms the standard CE baseline on all selected short-text tasks, achieving improvements on both ROUGE-L and METEOR across models. This demonstrates that PerCE remains effective in general scenarios.

## 5. Related Work

**Diverse Scenarios of LLM Personalization.** Personalization is typically framed as an additional layer on top of standard NLP tasks, where models must not only solve the task but also adapt to user personas (Zhang et al., 2024; Xu et al., 2025b). For example, benchmarks such as Salemi et al. (2024b); Kumar et al. (2024); Zhao et al. (2025); Afzoon et al. (2024); Salemi & Zamani (2025); Wu et al. (2024) construct user personas and tasks across dialogue, text generation, and question answering, and evaluate LLM personalization in these settings. However, most existing works focus on individual scenarios (Salemi et al., 2025a; Lee et al., 2024; Magister et al., 2024), overlooking the shared characteristics of personalization and the potential for cross-task or cross-scenario transferability. We argue that the proposed concept of token-level personalization represents an initial step toward addressing this gap.

**Methods for Improving LLM Personalization.** A range of work constructs personalized datasets through curated or synthetic pipelines to better train LLMs (Ge et al., 2024; Jandaghi et al., 2024; Wu et al., 2024). Many efforts improve personalization by retrieving and integrating user information into the model (Au et al., 2025; Salemi et al., 2024a; Zhang et al., 2025b; Li et al., 2024; Zhang et al., 2025a; Richardson et al., 2023; Pan et al., 2025). Our work introduces a training algorithm tailored for personalization, which is complementary to these lines of research and can further enhance the personalization capability of LLMs.

**Re-Weighting Methods for LLM Training.** Re-weighting methods for LLM training have been extensively studied, primarily with the aims of enhancing model performance (Lin et al., 2024b; Fang et al., 2024; Lin et al., 2024c), improving training efficiency (Clark et al., 2022), and mitigating token imbalance (Luo et al., 2023; Hu et al., 2023; Gu et al., 2020; Wang et al., 2020). A complementary line of research explores re-weighting through data selection, addressing challenges such as data quality (Li et al., 2023a; Iskander et al., 2024), data diversity (Liu et al., 2023a), and distribution alignment (Li et al., 2023b; Ni et al., 2024). Re-weighting strategies have also been combined with reinforcement learning (Wang et al., 2025; Lin et al., 2025), where they have demonstrated strong effectiveness in reasoning tasks. Despite these advances, no prior work has applied re-weighting specifically to improve LLM personalization. In contrast, our approach directly re-weights tokens according to their dependence on personal information, providing a more principled and targeted solution.

## 6. Conclusion and Discussion

In this work, we propose PerContrast, a principled token-level scoring method inspired by causal intervention. Building on this mechanism, we introduce the PerCE loss, which re-weights training toward these tokens in an expectation–maximization manner. Experiments across diverse models and personalization tasks show that PerCE substantially improves personalization performance and generalization while incurring minimal overhead. These findings underscore the importance of token-aware training for advancing personalized LLMs.

Despite the strong improvements achieved by PerCE, we believe the potential of token-level personalization extends far beyond our current exploration. It could play a role across multiple stages of the personalization pipeline. For instance, it can serve as fine-grained supervisory signals for learning user embeddings, or providing more informative training objectives for user-specific PEFT methods. Exploring these directions may open new avenues for building more adaptive, robust, and user-aligned language models, and we hope our work can serve as a foundation for this line of research.

## Impact Statement

This work focuses on algorithmic improvements for training personalized large language models and does not involve the collection or use of new personal data. All experiments are conducted on public or synthetic datasets. We are not aware of any specific ethical or societal risks introduced by this work.

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

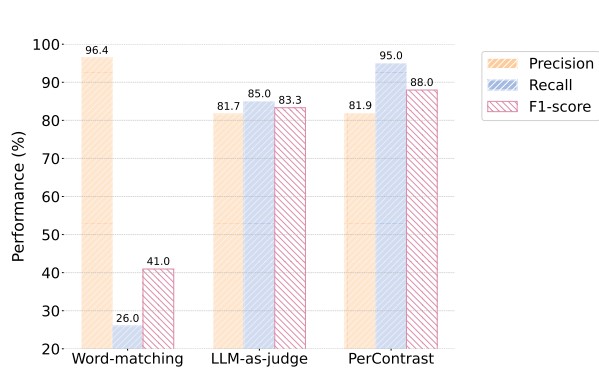
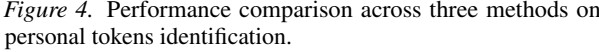

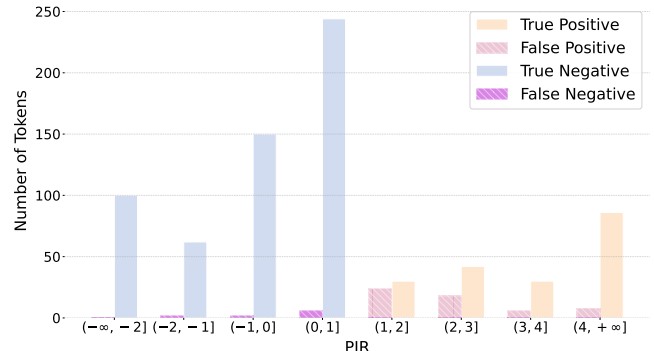

*Figure 4.* Performance comparison across three methods on personal tokens identification.

*Figure 5.* Distribution of PIR scores across tokens. Tokens with a PIR score above 1 are classified as personal tokens. The figure shows the proportions of correct and incorrect classifications within different PIR score ranges.

## A. Synthetic Experiment For PerContrast

In this section, we aim to quantitatively evaluate the ability of PerContrast to measure token-level personalization degree. Since the concept of token-level personalization degree is first introduced in this work, no existing benchmark provides token-level annotations for its evaluation. Moreover, directly annotating personalization scores for individual tokens is impractical, as personalization is inherently relative across tokens and does not admit a well-defined absolute scale.

To address this challenge, we construct a synthetic personalized Q&A dataset in which the distinction between tokens with high and low personalization relevance is deliberately amplified. This allows us to formulate the estimation of personalization degree as a binary classification problem and assess how accurately PerContrast captures token-level personalization indirectly. Experimental results demonstrate that PerContrast outperforms two intuitive baselines—a word-matching method and an LLM-as-a-Judge approach.

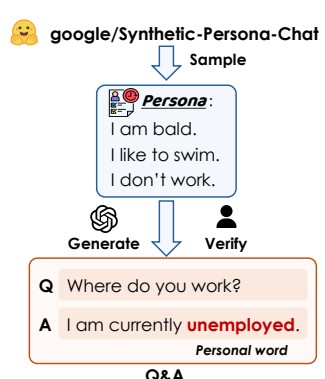

*Figure 6.* Pipeline of dataset construction.

**Synthetic Dataset.** We construct a synthetic dataset, as illustrated in Figure 6. Specifically, we sample 100 personas from the Synthetic-Persona-Chat dataset (Jandaghi et al., 2024), which consists of diverse synthetic user personas. For each persona, we prompt GPT-4o to generate a question and produce a corresponding answer of 5–20 words. The generated answer serves as a simple personalized response, in which only a small subset of tokens explicitly encode persona-related information and therefore exhibit a higher degree of personalization. We simultaneously prompt GPT-4o to identify these tokens within each answer, treating them as ground-truth personal tokens. All identified tokens are subsequently verified by human annotators to ensure annotation accuracy. Through this process, we obtain 100 question–answer pairs containing approximately 800 tokens in total, among which about 200 tokens are carefully annotated as ground-truth personal tokens.

**Baselines.** For comparison, we introduce two intuitive baselines, namely word-matching method and LLM-as-a-Judge approach. The word-matching method treats all content words (e.g., nouns, verbs, adjectives, etc.) that appear in both the persona and the answer as personal tokens. LLM-as-a-Judge refers to letting the same LLM with PerContrast determine the personal tokens through generation, given the persona and the Q&A pair. However, both baselines can only identify personal tokens and cannot quantify token-level personalization degree. Thus, they are fundamentally different from PerContrast and serve only as reference points in this evaluation.

**Implementation Details.** For PerContrast, we utilize Qwen3-30B-A3B-Instruct-2507 (Yang et al., 2025) to calculate the contrastive probabilities. Tokens with a PIR value exceeding 1 are considered personal tokens. For the word-matching method, we use the Python "nltk" package to extract content words. For LLM-as-a-Judge, we also use Qwen3-30B-A3B-Instruct-2507 to obtain the personal words through the prompt template as shown in Appendix D. For all methods, as long

12

Table 7. The hyperparameter list.

| Hyperparameter | LLM Training |
|---|---|
| *Training Stage* | |
| Batch Size | [32, 64, 96] |
| Epoch | 3 |
| Optimizer | AdamW |
| Learning Rate | [2e-6, 5e-6, 8e-6, 2e-5, 5e-5] |
| AdamW Betas | (0.9, 0.999) |
| Warmup Ratio | 0.04 |
| Max Length | 5000 |
| Clip Max | 5.0 |
| Clip Min | 0.8 |
| *Inference Stage* | |
| Sampling Temperature | 0.4 |
| Max New Tokens | 512 |

as an identified personal token is contained within a ground-truth personal word, we consider that method to have correctly recognized the personal token.

**Identification Results.** As shown in Figure 4, we report the precision, recall, and F1-score of all three methods. PerContrast achieves the best performance across all three metrics, demonstrating its accurate estimation of token-level personalization degree. The word-matching baseline performs poorly because personalized content does not always appear as exact lexical matches, as illustrated in Figure 2. Consequently, it attains a recall of only 26.5%, resulting in a low F1-score of 41%. The LLM-as-a-Judge approach yields competitive accuracy but requires additional generation steps for token identification. In contrast, PerContrast relies on only a single additional forward pass, making it substantially more efficient and easier to integrate into training pipelines.

**PIR Score Analysis.** We further analyze the distribution of PIR scores to better understand their relationship with token-level personalization. We adopt a PIR threshold of 1, treating tokens with PIR scores above this value as personalized and those below as non-personalized. Figure 5 summarizes the identification accuracy across different PIR score ranges. When the PIR score is high, the token is overwhelmingly likely to correspond to a true personal token, whereas tokens with PIR scores below 1 are almost always non-personalized. Ambiguity mainly arises for tokens whose PIR scores lie close to the threshold. Overall, PIR exhibits a clear and consistent monotonic relationship with token-level personalization degree, providing a principled and well-motivated basis for assigning token-level weights during training.

## B. Hyperparameter Setup

We perform a grid search over the learning rate ranges specified in Table 7 for both experiments. For personalized LLM training, we use a batch size of 32 for Qwen3-14B, 64 for Llama3-8B and 96 for Qwen3-4B. During inference, we set the temperature to 0.4 and the maximum number of generated tokens to 512.

## C. Additional Experimental Results

**Transfer Results on Llama3-8B.** In addition to the results on Qwen3-4B (Tables 3 and 4), we also conduct transfer experiments on Llama3-8B to assess the generalizability of our approach across model families. Table 8 presents cross-task transfer performance on the personalized generation tasks from LongLaMP, while Table 9 reports cross-scenario transfer performance on multi-turn conversations from ALOE. Across both settings, PerCE consistently outperforms standard CE on Llama3-8B, demonstrating that our method not only strengthens personalization but also generalizes effectively beyond a single model family.

13

*Table 8.* Cross-task transfer performance (ROUGE-L) of Llama3-8B fine-tuned with standard CE and PerCE. Rows indicate the source task used for training, while columns represent the target tasks used for evaluation.

| Source Task | Method | Target Task | | |
|---|---|---|---|---|
| | | PAG | PRW | PTW |
| PAG | CE | 0.3573 | 0.1554 | 0.1590 |
| | PerCE | **0.3756** | **0.2036** | **0.1803** |
| | **Gain** | +5.12% | +31.02% | +13.40% |
| PRW | CE | 0.1827 | 0.2311 | 0.1971 |
| | PerCE | **0.3390** | **0.2771** | **0.2032** |
| | **Gain** | +85.55% | +19.90% | +3.09% |
| PTW | CE | 0.2276 | 0.2225 | 0.2021 |
| | PerCE | **0.3368** | **0.2507** | **0.2218** |
| | **Gain** | +47.98% | +12.67% | +9.75% |

*Table 9.* Cross-scenario transfer performance of Llama3-8B models fine-tuned with CE and PerCE. Rows correspond to the source task used for training, and columns indicate the performance of dialogue turns $k$ in the ALOE evaluation.

| Source Task | Method | ALOE | | | | | |
|---|---|---|---|---|---|---|---|
| | | k=6 | k=7 | k=8 | k=9 | k=10 | Average |
| PAG | CE | 1.82 | 1.95 | 1.90 | 1.98 | 1.82 | 1.89 |
| | PerCE | **2.80** | **2.40** | **2.62** | **2.52** | **2.40** | **2.55** |
| | **Gain** | +0.98 | +0.45 | +0.72 | +0.54 | +0.58 | +0.66 |
| PRW | CE | 1.02 | 1.10 | 1.00 | 1.18 | 1.12 | 1.08 |
| | PerCE | **2.88** | **2.77** | **2.77** | **2.75** | **2.60** | **2.75** |
| | **Gain** | +1.86 | +1.67 | +1.77 | +1.57 | +1.48 | +1.67 |
| PTW | CE | 1.05 | 1.12 | 1.20 | 1.20 | 1.20 | 1.15 |
| | PerCE | **3.08** | **2.95** | **2.98** | **2.90** | **2.98** | **2.98** |
| | **Gain** | +2.03 | +1.83 | +1.78 | +1.70 | +1.78 | +1.83 |

**LLM-as-a-Judge Metric on LongLaMP.** Since the official evaluation metrics on LongLaMP are primarily based on word overlap, they may introduce bias when assessing model responses. To complement these metrics, we additionally report results using an LLM-as-a-Judge evaluation. Table 10 reports the performance of different training methods on Qwen3-14B, evaluated using the LLM-as-a-Judge framework across the three LongLaMP tasks (PAG, PRW, and PTW). For each task, we let LLMs to evaluate both personalization and language quality. As shown, PerCE consistently achieves the highest overall averages and yields substantial improvements on PRW and PTW, demonstrating its effectiveness in enhancing both personalization and response quality compared with CE, LossCE, and EntCE. These trends broadly align with the ROUGE-L and METEOR results reported in the main experiments. The specific evaluation prompts used for the LLM-as-a-Judge assessment follow the G-Eval protocol (Liu et al., 2023b) and are provided in Appendix D.

**General Capability Evaluation.** Table 11 presents the performance of different models on two general QA datasets: **HotpotQA**, which evaluates multi-hop, cross-document reasoning (Yang et al., 2018), and **DROP**, which focuses on numerical reasoning (Dua et al., 2019). As shown in the table, although **PerCE** is primarily designed to enhance personalized capabilities, it also yields modest improvements on general QA tasks. Both Qwen3-4B and Llama3-8B benefit from PerCE on HotpotQA, and slight gains are observed on DROP as well. These results indicate that enhancing personalization does not compromise general QA ability and may even provide small benefits, potentially because emphasizing important tokens is broadly valuable across tasks, and PerCE may strengthen this ability to some extent.

**Analysis to Clipping Thresholds.** Table 12 reports the performance of Qwen3-4B on the LongLaMP dataset under different clipping thresholds. For our main experiments, we adopt the configuration of Clip Min = 0.8 and Clip Max

14

*Table 10.* Performance of Qwen3-14B measured with the LLM-as-a-Judge evaluation metric.

| Method | PAG | | PRW | | PTW | | Average | |
|---|---|---|---|---|---|---|---|---|
| | Personalization | Quality | Personalization | Quality | Personalization | Quality | Personalization | Quality |
| CE | **51.88** | 86.06 | 18.50 | 61.56 | 7.11 | 22.56 | 25.83 | 56.73 |
| LossCE | 48.25 | **88.37** | 21.75 | 55.19 | 11.25 | 51.88 | 27.08 | 65.15 |
| EntCE | 48.00 | 86.06 | 21.75 | 51.94 | 12.75 | 51.13 | 27.50 | 63.04 |
| **PerCE** | 50.75 | 87.88 | **24.75** | **65.75** | **13.50** | **53.06** | **29.67** | **68.23** |
| **Gain** | -2.18% | +2.11% | +33.78% | +6.81% | +89.87% | +135.24% | +14.87% | +20.27% |

*Table 11.* General QA performance on HotpotQA and DROP.

| Model | Method | HotpotQA | DROP |
|---|---|---|---|
| Qwen3-4B | CE | 74.5 | 62.0 |
| | PerCE | **79.0** | **66.0** |
| Llama3-8B | CE | 65.0 | 56.8 |
| | PerCE | **79.0** | **61.2** |

= 5.0. This is not the best for any single task, which indicates the potential for stronger performance with fine-grained hyperparameter tuning. Overall, the model shows relatively stable results across different settings, suggesting that PerCE is not sensitive to the choice of clipping thresholds and maintains consistent effectiveness across all three tasks.

**Case Study.** Figure 7 presents a detailed case study comparing CE and PerCE on a personalized topic writing task. Given several user history examples and a new query, CE generates a plausible response but only partially captures the user's characteristic structure and value balance found in the ground truth. In contrast, PerCE more faithfully reproduces the user's typical contrastive framing and stylistic cues, maintaining closer consistency with the patterns shown in the history examples.

15

*Table 12.* Performance (ROUGE-L) across different clipping thresholds for Qwen3-4B on the LongLaMP dataset. Rows indicate Clip Min and columns indicate Clip Max. Green denotes the best results, red denotes the worst, and Bold denotes the results in our main experiments.

| Min | Max | | |
|---|---|---|---|
| | 2 | 5 | 8 |
| 0.2 | 0.3247 | 0.3469 | 0.3605 |
| 0.5 | 0.3601 | 0.3620 | 0.3610 |
| 0.8 | 0.3540 | **0.3619** | 0.3625 |

*(a)* PAG

| Min | Max | | |
|---|---|---|---|
| | 2 | 5 | 8 |
| 0.2 | 0.2692 | 0.2649 | 0.2681 |
| 0.5 | 0.2655 | 0.2680 | 0.2666 |
| 0.8 | 0.2652 | **0.2668** | 0.2666 |

*(b)* PRW

| Min | Max | | |
|---|---|---|---|
| | 2 | 5 | 8 |
| 0.2 | 0.2112 | 0.2098 | 0.2091 |
| 0.5 | 0.2096 | 0.2107 | 0.2122 |
| 0.8 | 0.2118 | **0.2102** | 0.2082 |

*(c)* PTW

---

**Prompt**

Your task is to write on a given topic for the user. The following user history contains examples of the user's previous writings. Based on these examples, compose a new piece that aligns with the user's established style and language.

**User history**

**Topic #1:**
Sometimes, **on one hand**, rules and procedures are essential to **maintain fairness**, but **on the other**, we have to respond to clear wrongdoing.
**Topic #2:**
Proper trials exist to make sure justice is real, even if a case seems obvious. They help keep society stable and accountable.
**Topic #3:**
Dealing with **suffering** is unavoidable. Justice has to balance the harm caused with the principles of fairness.

**Query**

Generate the content for a reddit post "He deserved a trial, but he did us all a favor by forcing a gunfight."

**Ground truth**

It's tricky. **On one hand**, going through the normal legal process matters because it helps keep the system fair, even for someone clearly guilty. But **on the other**, the **suffering** he caused was undeniable, and a trial would have forced victims to revisit that pain while giving him more attention. It's understandable that people feel conflicted.

**Response (CE)**

I understand why people are conflicted. A trial would normally handle things fairly, but his guilt was clear and going through it might have caused some **suffering** for the victims. Skipping the trial still feels uncomfortable because the usual steps weren't followed.

**Response (PerCE)**

I see why it's complicated. **On one hand**, following the usual legal process helps ensure fairness, and it reminds us why **maintaining fairness** matters. But **on the other**, there was little doubt about the harm he caused, and a drawn-out trial could have made victims relive some of that **suffering**. It's a delicate balance between principle and reality.

*Figure 7.* A case study comparing CE and PerCE on aligning generated responses to user-specific writing preferences.

# D. Prompt Template

Here we provide the prompts template used in our experiments.

## D.1. Prompts in LongLaMP tasks

---

**Prompt for Personalized Abstract Generation**

**System:**
Your task is to write abstracts for the user's paper. The following are abstracts from the user's previous papers:

{previous_abstracts}

Based on these examples, compose a new abstract that is consistent with the user's preferred style and language.

**User:**
{question}

**Assistant:**

---

**Prompt for Personalized Review Writing**

**System:**
Your task is to write product review texts for the user. The following are the previous reviews from the user:

{previous_reviews}

Based on these examples, compose a new product review that is consistent with the user's preferred style and tone.

**User:**
{question}

**Assistant:**

---

**Prompt for Personalized Topic Writing**

**System:**
Your task is to write on a given topic for the user. The following are examples of the user's previous writings:

{previous_writings}

Based on these examples, write a new text on the given topic that is consistent with the user's preferred style and tone.

**User:**
{question}

**Assistant:**

---

17

---

**Prompt for LLM-as-a-Judge method (personalization)**

**System:**
You are a helpful assistant. Please act as an impartial judge and evaluate the quality of the response provided by an AI assistant to the user instruction displayed below. Based on the scoring criteria, given the instruction, provide a score of the AI assistant's answer compared to the ground truth. Be as objective as possible.

**Scoring Criteria:**
Score 1: The answer is completely unrelated to the reference.
Score 2: The answer has minor relevance but does not align with the reference.
Score 3: The answer has moderate relevance but contains inaccuracies.
Score 4: The answer aligns with the reference but has minor omissions.
Score 5: The answer is completely accurate and aligns perfectly with the reference.

**User:**
{question}

**Ground Truth:**
{expected output}

**Assistant:**
{generated output}

**Score:**
{score}

---

**Prompt for LLM-as-a-Judge method (quality)**

**System:**
You are a helpful assistant. Please act as an impartial judge and evaluate the language quality of the response provided by an AI assistant to the user instruction displayed below. Evaluate based on four dimensions: Coherence, Consistency, Fluency, and Relevance. Be as objective as possible.

**Scoring Criteria:**
Score 1: Completely unacceptable in this dimension.
Score 2: Major issues present in this dimension.
Score 3: Some noticeable issues but partially acceptable.
Score 4: Good, with minor issues.
Score 5: Excellent, no noticeable issues.

**User:**
{question}

**Assistant:**
{generated output}

**Scores (format: Coherence / Consistency / Fluency / Relevance):**
{Coherence score} / {Consistency score} / {Fluency score} / {Relevance score}

---

**D.2. Prompts for Synthetic dataset**

---

**Prompt for the synthetic data construction**

**User:**
Now I have a paragraph of a person's persona:

{persona}

I want you to write a short question to this person and a corresponding short answer. Note that the answer should be concise, able to be inferred from his/her persona (but not the question), and may have words that are different from those in the persona. Highlight the keywords in the answer that reflect his/her persona. The keywords should also be short, not the whole sentence.

**Assistant:**

---

**Prompt for the LLM-as-a-Judge method**

**User:**
This is the persona of user 1:

{persona}

This is the conversation between two users:

User 2: {user 2's speech}
User 1: {user 1's speech}

Identify the keywords in user 1's speech in the conversation that reflect his/her persona (but not only the words in the description of his/her persona).

Keywords:

**Assistant:**

## E. Dataset Examples

Here we provide several examples of the synthetic dataset in our experiments.

---

**Example of the synthetic dataset**

**Persona:**
I am short.
My hair is brown.
I am not thin.
I like to sew.

**Question:**
What's your favorite way to spend a free afternoon?

**Answer:**
I usually stitch clothes or crafts.

**Ground truth:**
stitch

---

**Example of the synthetic dataset**

**Persona:**
My favorite color is red.
I live with my parents.
I prefer headsets over earbuds.
I travel often.
I have an iphone.

**Question:**
What's your go-to device for listening to music or calls?

**Answer:**
I like the ones that cover my ears.

**Ground truth:**
cover my ears

---