# OpenReview forum: "Rethinking Personalization in Large Language Models at the Token Level"
_ICML.cc/2026/Conference — ICML 2026 regular_

### Official Review · Reviewer_dBxX · 2026-03-03

**Soundness:** 3
**Presentation:** 3
**Significance:** 3
**Originality:** 2
**Overall Recommendation:** 4
**Confidence:** 3

**Summary:**

Not every word in a model’s response matters equally when it comes to personalization—some tokens carry more user-specific weight and deserve extra attention during training. However, accurately estimating the personalization degree of these tokens has been a persistent challenge. To address this, the research team proposed PerContrast, a method that identifies personalized tokens through causal intervention, and developed the PerCE loss function to adaptively upweight these tokens during training, thereby effectively enhancing the model's personalization capability.

**Compliance With Llm Reviewing Policy:**

Affirmed.

**Final Justification:**

The rebuttal addressed my concern, I raised my score.

**Key Questions For Authors:**

1. How do you get the personalized information (in other words, personalized tokens) from the training data?

2. see weakness

**Limitations:**

yes

**Strengths And Weaknesses:**

Strength

This work target on the important research topic of personalized LLM, which is important as the fast development of LLMs’ application. They conduct the first token-level analysis of personalization and introduce PerContrast, an efficient selfcontrast
method that quantifies each token’s contribution to personalization with causality theoretical guarantee. They also develop the PerCE loss, which enhances the model’s personalization capabilities in an expectation–maximization (EM) manner

Weak：

1.The method relies on extracting personalized information from training corpora, which is often unavailable in real-world scenarios without manual labeling. The paper lacks a thorough discussion on how labeling accuracy impacts the overall performance.

2.Insufficient Benchmarking: The evaluation fails to include recent state-of-the-art personalized benchmarks, specifically:

[1]·  PREFEVAL:DO LLMS RECOGNIZE YOUR PREFERENCES? EVALUATING PERSONALIZED PREFERENCE FOLLOWING IN LLMS, ICLR 2025

[2]Personalized Safety in LLMs: A Benchmark and A Planning-Based Agent Approach, Neurips 2025

3.Comparisons are currently limited to token-level baselines. To demonstrate true effectiveness, the model should be benchmarked against established personalized methods.

4. Typos such as “causal effect To evaluate its” Line 100 Page 2.

---

> ### Author Rebuttal · Authors · 2026-03-31
>
> We appreciate your highlighting the **novelty of our approach, particularly the first token-level formulation and its causal grounding.** The concerns—primarily on evaluation scope and baselines—are addressed below with additional clarifications and experiments.
>
> ---
>
> W1. Availability of labeled personalization information
>
> We clarify that personalized information is typically available in real-world personalization settings, rather than requiring manual annotation.
>
> In personalization tasks, models are expected to infer user preferences from **historical interactions**. These signals are naturally accumulated during user usage and are commonly incorporated into the model input through **constructed user profiles or retrieved user histories**. In other words, personalization information is not manually labeled but **automatically generated and injected into the prompt** as part of the system pipeline.
>
> Personalization benchmarks generally follow the same setting. They either provide full user histories [1, 2] or supply pre-construct user preferences or profiles fo the model to condition on (e.g., the benchmarks you mentioned in W2 [3, 4]).
>
> ---
>
> W2. Insufficient Benchmarking
>
> We provide additional experiments on these benchmarks (all on LLaMA-8B).
>
> PrefEval [3] is a benchmark for long-context, multi-turn personalized dialogue, evaluating whether models can maintain alignment with user preferences over extended conversations. Following the original setup, we train on 10-turn dialogues and evaluate on conversations ranging from 2 to 70 turns. As shown in below Table, **PerCE achieves consistent and significant improvements across all dialogue lengths.**
>
> |PrefEval|2|5|10|30|50|70|ave|
> |-|-|-|-|-|-|-|-|
> |CE|83.52|82.42|75.27|76.37|75.82|69.23|77.11|
> |PerCE|85.71|85.16|82.42|78.57|77.47|80.77|81.68|
>
> PENGUIN [4] (personalized safety) has no training set, so we evaluate in a transfer setting using a LongLaMP-trained LLaMA-8B checkpoint. PerCE shows **stronger cross-scenario generalization**:
>
> |PENGUIN|Career|Education|Financial|Health|Life|Relationship|Social|ave|
> |-|-|-|-|-|-|- |-|-|
> | CE | 2.35 | 2.15 | 2.28 | 2.15 | 2.30 | 2.07 | 2.13 | 2.21 |
> | PerCE | 3.75 | 3.82 | 3.72 | 4.03 | 3.71 | 3.89 | 3.72 | 3.8 |
>
> ---
>
> W3. Comparisons are currently limited to token-level baselines. To demonstrate true effectiveness, the model should be benchmarked against established personalized methods
>
> We would like to clarify that our method introduces a training loss that is **orthogonal and complementary to many existing personalization approaches.**
>
> For example, many inference-time methods focus on constructing better user personas to ground the model in user preferences [2,5]. Our current setup adopts a standard RAG-based approach and can be directly combined with more advanced retrieval or profiling techniques.
>
> Many training-time methods encode user personas into model parameters via PEFT or learned embeddings [6,7]. Our method is also compatible with such approaches, as it only modifies the training loss and does not impose constraints on the model architecture or parameterization. To provide a preliminary validation, we combine PerCE with LoRA and evaluate on LongLaMP. The results show that PerCE consistently improves performance over standard CE training under the same LoRA setting, confirming its complementary nature.
>
> | LongLamp-LoRA | a | p | t |
> | --- | --- | --- | --- |
> | CE | 36.00 / 31.08 | 19.39 / 16.89 | 16.01 / 13.59 |
> | PerCE | **36.56** / **31.26** | **21.15** / **18.26** | **18.91** / **18.05** |
>
> ---
> W4. Typos in Page 2.
>
> Thank you for pointing this out. We will correct them.
>
> ---
> Q1. Extraction of personalized information from training data
>
> As discussed in W1, personalized information is typically available in real-world personalization settings and is reflected in training data.
>
> In practice, training data usually follow the same paradigm: they either provide user histories or user personas. During inference, the system retrieves relevant user history or constructs a user profile and injects it into the model input. Therefore, personalized information is naturally embedded in the input and can be **automatically identified without manual labeling**.
>
> ---
> **Reference**
>
> [1] LongLaMP: A Benchmark for Personalized Long-form Text Generation, ACL2025
>
> [2] LaMP-QA: A Benchmark for Personalized Long-form Question Answering, ACL2025
>
> [3] Do LLMs Recognize Your Preferences? Evaluating Personalized Preference Following in LLMs, ICLR2025
>
> [4] Personalized Safety in LLMs: A Benchmark and A Planning-Based Agent Approach, NeurIPS 2025
>
> [5] SeCom: On Memory Construction and Retrieval for Personalized Conversational Agents, ICLR2025
>
> [6] LLMs + Persona-Plug = Personalized LLMs, ACL2025
>
> [7] Democratizing Large Language Models via Personalized Parameter-Efficient Fine-tuning, EMNLP2024
>
> ---
> Thanks again, and we hope the response can fully address your concern! We would appreciate if you could reconsider the ratings.

---

> > ### Author Rebuttal · Reviewer_dBxX · 2026-04-03
> >
> > Regarding Week 1, the response is unconvincing. I agree that "personalized information is typically available in real-world personalization settings, rather than requiring manual annotation." However, the paper's core claim—that "some tokens carry more user-specific weight and deserve extra attention during training"—needs stronger support. I would like to see more rigorous proof and analysis to confirm that the detected tokens are indeed related to personalization.

---

> > > ### Author Response · Authors · 2026-04-05
> > >
> > > Thank you for your feedback. We sincerely appreciate your time and effort in evaluating our work.
> > >
> > > Your previous comments mainly focused on basic personalization settings (W1 & Q1), insufficient benchmarks (W2), and compared baselines (W3). We are glad that our earlier responses have helped clarify these aspects. Below, we address your new concern regarding the need for “*rigorous proof and analysis to confirm that the detected tokens are indeed related to personalization”*, from both theoretical and empirical perspectives.
> > >
> > >
> > > **Theoretical Analysis**
> > >
> > > In Section 2.2, we provide a rigorous theoretical analysis showing that Personal Influence Ratio (PIR) is equivalent to the causal effect from user persona to each token (Theorem 2.3). In our method, token importance is determined based on PIR. Therefore, **tokens with higher PIR correspond to those with stronger causal influence from the user persona, providing a principled basis for identifying personalization-relevant tokens.**
> > >
> > >
> > > **Empirical Evidence**
> > >
> > > In Appendix A, we conduct synthetic experiments to provide an intuitive demonstration of the relationship between detected tokens and personalization. Specifically, we construct a synthetic personalized Q&A dataset in which the distinction between tokens with high and low personalization relevance is deliberately amplified. Figure 4 shows that PIR can accurately identify tokens with high personalization relevance. Figure 5 further presents fine-grained results, exhibiting a clear and consistent monotonic relationship between PIR and token-level personalization degree.
> > >
> > >
> > > **Summary**
> > >
> > > Overall, **both our theoretical analysis and empirical evidence consistently support that PIR effectively captures token-level personalization influence, and that the detected tokens are indeed closely related to personalization.**
> > >
> > > Based on this, a natural implication is that tokens with higher personalization relevance should receive greater emphasis during training. This motivates our proposed PerCE method. Empirically, PerCE achieves consistent and significant improvements across a wide range of benchmarks, including LongLaMP, LaMP, ALOE, PrefEval, and PENGUIN, further validating this intuition.
> > >
> > > ---
> > >
> > > Thank you again for your thoughtful feedback. We hope our response helps clarify your concerns, and we would greatly appreciate it if you would consider updating your assessment accordingly.

---

### Official Review · Reviewer_AvM4 · 2026-03-11

**Soundness:** 3
**Presentation:** 3
**Significance:** 3
**Originality:** 3
**Overall Recommendation:** 4
**Confidence:** 3

**Summary:**

This paper explores LLM personalization from a lexical-level perspective, rather than the traditional sequence-level approach. The authors propose the PerContrast method, which calculates the Personal Influence Ratio (PIR) of each generated token using causal intervention (masking user identity). Based on this metric, they introduce the PerCE loss function. This function uses an online guidance mechanism similar to the Expectation-Maximization (EM) algorithm to adaptively adjust the weights of tokens based on their relevance to personalization during training. The proposed method shows improved performance in long text personalization generation tasks.

**Compliance With Llm Reviewing Policy:**

Affirmed.

**Key Questions For Authors:**

Q1. Is it feasible to estimate the PIR directly through internal model signals (e.g., attention weights mapping to the persona context) or gradient approximations, thereby eliminating the computational overhead of the extra forward pass?

Q2. If a user's personalization profile is exceptionally long (e.g., dozens of retrieved documents), wouldn't masking the entire history in PerContrast cause severe distribution shifts, potentially distorting the PIR estimation?

Q3. How sensitive is the EM-like bootstrapping process to the initial capabilities of the base model? Is there a risk of confirmation bias where the model amplifies its own hallucinations if the initial PIR estimates are noisy?

Q4. How does the PerCE loss affect tokens that have low PIR but are critical for grammatical correctness and fluency? Did you observe any degradation in general language modeling capabilities?

**Limitations:**

yes

**Strengths And Weaknesses:**

This paper refines the granularity of personalized research in Language Learning Models (LLM) to the lexical level and introduces the Potential Outcomes Framework (POFTA) to explain Personal Influence Rate. Its proposed bootstrapping mechanism is similar to the Expectation-Maximization (EM) algorithm, allowing training without lexical-level preference labeling, thus ensuring theoretical consistency.

However, this method has significant limitations in engineering implementation: calculating PIR requires additional forward propagation, significantly increasing the computational cost of training. Furthermore, current experimental scenarios are mainly limited to standard text generation; the risk of distribution shift due to masking the entire historical context in long contexts or multi-turn dialogues has not been fully validated. Despite these computational trade-offs, this method still provides a novel perspective and a solid theoretical framework. I would be inclined to accept this paper if the authors could answer the following question.

---

> ### Author Rebuttal · Authors · 2026-03-31
>
> We appreciate your recognition of our work’s **novel perspective and solid theoretical framework**. The concerns regarding computational overhead and evaluation scope are valuable. We address them below with additional experiments and clarifications.
>
> ---
> W1. Computational overhead of additional forward pass
>
> We respectfully disagree. While an additional forward pass is introduced, the overhead is modest:
>
> 1. **Backward dominates cost**, so the relative overhead of one extra forward is limited.
> 2. **Shorter forward.** The auxiliary pass excludes persona tokens, reducing sequence length and cost.
>
> Table 1 shows per-step training time (s) for Qwen3-4B on 4×A100 (batch=3, grad acc=8). The overhead is only **15.3%**, which is acceptable.
>
> | LongLaMP | PAG | PRW | PTW | AVE |
> | --- | --- | --- | --- | --- |
> | CE | 59.5 | 59.1  | 61.9 | 60.2 |
> | PerCE | 67.4 | 71.4 | 69.5 | 69.4 |
>
> Our implementation is currently sequential without engineering optimization; the two forwards can be parallelized, further reducing cost. This is also consistent with feedback from reviewer sX2m noting the low practical overhead.
>
> ---
> W2. Distribution shift in long-context and multi-turn settings
>
> We address this with additional experiments.
>
> **PEToolBench [1] (long context) .**
>
> Evaluates personalization from long histories (~50 turns) with noisy signals. PerCE shows consistent gains.
>
> | PEToolLM (qwen4b) | preferenced-only | rating-integrated | chronological | ave |
> |-|-|-|-|-|
> | CE | 76.9 / 92.0 | 76.9 / 93.0  | 70.6 / **92.5** | 74.8 / 92.5 |
> | PerCE | **78.0** / **93.4** | **82.2** / **94.2** | **74.0** / 92.0 | **78.1** / **93.2** |
>
> **PrefEval [2] (multi-turn).**
>
> Training on 10 turns, testing up to 70.
>
> | PrefEval (llama8b) | 2 | 5 | 10 | 30 | 50 | 70 | ave |
> |-|-|-|-|-|-|-|-|
> | CE | 83.52 | 82.42 | 75.27 | 76.37 | 75.82 | 69.23 | 77.11 |
> | PerCE | **85.71** | **85.16** | **82.42** | **78.57** | **77.47** | **80.77** | **81.68** |
>
> These results demonstrate robustness to long contexts and potential distribution shifts.
>
> [1] PEToolLLM: Towards Personalized Tool Learning in Large Language Models, ACL2025
>
> [2] Do LLMs Recognize Your Preferences? Evaluating Personalized Preference Following in LLMs, ICLR 2025
>
> ---
> Q1. Estimating PIR from internal model signals
>
> This is an interesting direction, but we remain cautious:
>
> 1. **Lack of reliable counterfactual estimation.** Internal signals are computed with persona present and cannot reliably approximate the no-persona counterfactual, while our extra forward explicitly constructs it.
>
> 2. **Low overhead.** As shown in W1, the cost of extra forward pass is acceptable in practical settings.
> 3. **Engineering compatibility.** Internal-signal methods require modifying computation graphs and storing activations. This not only introduces overhead but also reduces compatibility with existing training frameworks, potentially limiting practical adoption.
>
> Thus, explicit counterfactual estimation is more reliable and practical.
>
> ---
> Q2. Long personalization profiles
>
> As shown in W2, PEToolBench results demonstrate strong performance even with long and noisy histories.
>
> ---
> Q3. Sensitivity of EM bootstrapping
>
> We find the method **not sensitive to initialization**. Even with Qwen3-4B, which is relatively small by current standards, PerCE performs well across LongLaMP, LaMP, and PEToolBench.
>
> From an algorithmic perspective, two designs reduce the risk of confirmation bias:
>
> 1. **PIR Clipping.**
>     We apply upper and lower bounds to PIR when using it as training weights. This prevents extreme or noisy estimates from disproportionately influencing the optimization process.
>
> 2. **Alignment with token prediction accuracy.**
>     PIR estimation is strongly correlated with the model’s next-token prediction capability on the task. Even if the initial estimates are somewhat noisy, the model progressively improves its predictions during training, which in turn refines the PIR estimates and stabilizes the bootstrapping process.
>
> Overall, both empirical results and design considerations indicate that the method is robust to initialization and does not exhibit significant confirmation bias in practice.
>
> ---
> Q4. Impact of PerCE on low-PIR tokens and general LM performance
>
> Tokens with low PIR are assigned smaller weights, which we consider appropriate for two reasons.
> 1. Modern language models already exhibit strong grammatical correctness and fluency, so down-weighting these tokens does not harm these abilities but allows the model to focus more on personalization, the primary bottleneck.
> 2. This reweighting helps prevent overfitting to the current task and better captures personalization-specific signals.
>
> Empirically, we observe **no degradation** in general language modeling capabilities. As shown in Table 11 (Appendix), PerCE achieves comparable or slightly better performance than standard CE on general QA tasks.
>
> ---
> Thanks again, and we hope the response can fully address your concern!

---

### Official Review · Reviewer_sX2m · 2026-03-13

**Soundness:** 3
**Presentation:** 3
**Significance:** 2
**Originality:** 2
**Overall Recommendation:** 4
**Confidence:** 4

**Summary:**

This paper focuses on token-level personalization in LLM generation, claiming that not all generated tokens depend equally on user information. This paper proposes PerContrast to measure the degree to which each token is influenced by persona, measured by the probability difference with and without persona. Using this signal, they construct PerCE, a token-weighted training loss that emphasizes more personalized tokens during fine-tuning.

**Compliance With Llm Reviewing Policy:**

Affirmed.

**Final Justification:**

The authors’ rebuttal addressed most of my main concerns, especially through the additional experiments on model generalization and the clearer explanation of the causal effect analysis. These additions significantly improved my confidence in the soundness of the paper and clarified important aspects that were previously insufficiently justified. Overall, while I still think there is some room for improvement in presentation and clarity, the paper’s methodology and contribution are now much better supported, and the rebuttal has positively changed my evaluation. Based on this, I have decided to increase my score to 4.

**Key Questions For Authors:**

Q1: Could the authors provide a more rigorous justification for this interpretation, or discuss under what assumptions the probability difference can reliably capture the influence of personalization?

Q2: Could the authors discuss how well the proposed method generalizes to larger models, different personalization settings, or other personalized generation tasks beyond LongLaMP?

**Limitations:**

Yes

**Strengths And Weaknesses:**

S1: This paper is well-motivated and easy to follow.

S2: The PerContrast method is relatively lightweight, requiring only one additional forward computation to remove the persona, without modifying the model structure. Therefore, it can be directly applied to existing personalized LLM training pipelines, resulting in low practical cost.

S3: On personalized generation tasks such as LongLaMP, PerCE achieves stable improvements over standard CE and some token weighting baselines, indicating its effectiveness in practice.

W1: The overall method is mainly based on simple probability difference estimation and loss reweighting, with relatively direct technical contributions. Compared to some recent personalization or token importance learning methods, its novelty is limited.

W2: The paper interprets PIR as a causal effect proxy of the persona on the token, but this interpretation remains largely intuitive, lacking rigorous theoretical analysis to explain under what conditions this metric can reliably reflect the influence of personalization.

W3: The experiments mainly focused on the LongLaMP benchmark and three LLM model sizes (Qwen and Llama), lacking validation in scenarios with more and larger models or more personalizations. Therefore, the generalization ability of the method still needs further proof.

---

> ### Author Rebuttal · Authors · 2026-03-31
>
> We appreciate your recognition of our work’s **effectiveness and low practical cost.** Some concerns may stem from misunderstandings, which we clarify below with additional explanations and experiments.
>
> ---
>
> W1: Novelty concern
>
> We respectfully disagree with the concern regarding novelty. As also noted by other reviewers, novelty is highlighted as a key strength of our work. To avoid misunderstanding, we would like to clarify that our contributions go beyond simple reweighting and introduce **fundamentally new perspectives**:
>
> 1. **A token-level formulation of personalization.**
>
>     Existing personalization methods operate at the sequence or profile level. In contrast, we are the first to formulate personalization at the **token level**, enabling fine-grained analysis of how individual tokens reflect and contribute to user preferences.
>
> 2. **An online bootstrap training framework.**
>
>     We propose an **EM-style online training framework with a self-contrastive estimation mechanism**, which dynamically updates estimated PIR during training rather than relying on static weights. This differs fundamentally from prior token importance methods.
>
>
> Overall, our work introduces a **new perspective and training paradigm** for personalization in LLMs.
>
> ---
>
> W2. Lack of rigorous justification for PIR as a causal effect
>
> We respectfully clarify that a formal theoretical analysis is provided in Section 2.2. Specifically, under a causal framework with causal graph shown in Figure 3, we model persona as the treatment,  input query $x$ as confounders, and define token-level potential outcomes. Under standard assumptions in causal community like **No Interference and** **Unconfoundedness,** we show that the token-level causal effect is identifiable and is **exactly equivalent to PIR** (Theorem 2.3).
>
> Notably, these assumptions are reasonable in our setting: (i) given $(x, y_{<i})$, next-token generation of one sample will not affect generation of other samples, satisfying no interference assumption; (ii) due to the next token generation process only depends on input query $x$ and persona $p$, there is no other issue affect next token generation, satisfying unconfoundedness assumption. Therefore, PIR is not merely an intuitive proxy, but a **theoretically grounded causal effect estimator** under clearly stated conditions.
>
> ---
>
> W3. Limited evaluation on larger models and diverse benchmarks
>
> Thank you for this suggestion. We provide additional results on **larger models** and **more benchmarks**.
>
> **Larger Models**
>
> In our main experiments, we evaluate models ranging from 4B to 14B, which already covers a broader range than most related works [1–3]. To further demonstrate scalability, we additionally evaluate Qwen3-30B on LongLaMP.
>
> | Qwen30B | a | p | t |
> | --- | --- | --- | --- |
> | CE | **0.3871** / 0.3425 | 0.2431 / 0.2574 | 0.1907 / 0.1842 |
> | PerCE | 0.3794 / **0.3561** | **0.2869** / **0.2743** | **0.2435** / **0.2296** |
>
> We observe consistent improvements on most metrics, indicating that PerCE scales effectively to larger models.
>
> **More Personalization Benchmark**
>
> In the paper, we evaluate on three benchmark: LongLaMP, LaMP, and ALOE. To further demonstrate generalization, we include additional results on **PEToolBench [4]** (personalized tool use) and **PrefEval [5]** (multi-turn personalized dialogue).
>
> | PEToolLM (qwen4b) | preferenced-only | rating-integrated | chronological | ave |
> | --- | --- | --- | --- | --- |
> | CE | 76.9 / 92.0 | 76.9 / 93.0  | 70.6 / **92.5** | 74.8 / 92.5 |
> | PerCE | **78.0** / **93.4** | **82.2** / **94.2** | **74.0** / 92.0 | **78.1** / **93.2** |
>
> | PrefEval (llama8b) | 2 | 5 | 10 | 30 | 50 | 70 | ave |
> | - | - | - | - | - | - | - | - |
> | CE | 83.52 | 82.42 | 75.27 | 76.37 | 75.82 | 69.23 | 77.11 |
> | PerCE | **85.71** | **85.16** | **82.42** | **78.57** | **77.47** | **80.77** | **81.68** |
>
> PerCE shows consistent gains across all settings, demonstrating strong generalization.
>
> [1] LLMs + Persona-Plug = Personalized LLMs, ACL2025
>
> [2] SeCom: On Memory Construction and Retrieval for Personalized Conversational Agents, ICLR2025
>
> [3] RHO-1: Not All Tokens Are What You Need, NeurIPS 2024
>
> [4] PEToolLLM: Towards Personalized Tool Learning in Large Language Models, ACL2025
>
> [5] Do LLMs Recognize Your Preferences? Evaluating Personalized Preference Following in LLMs, ICLR 2025
>
> ---
>
> Q1. Assumptions and justification for causal interpretation
>
> As in W2, we provide a theoretical justification in Section 2.2. Under standard causal assumptions, we show that the probability difference is **equivalent to the token-level causal effect**, thereby providing a principled interpretation of PIR.
>
> ---
>
> Q2. Generalization to larger models and diverse settings
>
> As discussed in W3, we provide additional results on larger models and more benchmarks.
>
> ---
>
> Thanks again, and we hope the response can fully address your concern! We would appreciate if you could reconsider the ratings.

---

> > ### Author Rebuttal · Reviewer_sX2m · 2026-04-03
> >
> > The authors have partially addressed my concerns. I appreciate their efforts and the improvements made, and I have decided to increase my score by one point.

---

> > > ### Author Response · Authors · 2026-04-05
> > >
> > > Thank you for your positive feedback and for increasing your score. We sincerely appreciate your time and effort in evaluating our work.
> > >
> > > Due to there are no concerns raised in the reply, to ensure clarity, we further summarize the concerns you raised initially and how we have addressed them:
> > >
> > > - **W1 (Novelty):** We further clarified the novelty of our method compared to existing work on personalization and token importance learning. In particular, we introduce (i) the *first token-level formulation of personalization*, which differs fundamentally from prior sequence- or profile-level approaches, and (ii) an *EM-style online bootstrap framework* that dynamically estimates and updates token-level personalization signals during training. Together, these contributions establish a new perspective and training paradigm for personalization in LLMs.
> > > - **W2 / Q1 (Causal interpretation of PIR):** We provided a rigorous theoretical analysis in Section 2.2. Under standard causal assumptions (e.g., no interference and unconfoundedness), we show that the probability difference (PIR) is identifiable and equivalent to the token-level causal effect (Theorem 2.3), thereby grounding PIR in a principled causal framework.
> > > - **W3 / Q2 (Experimental generalization):** We extended our evaluation to include (i) a larger model (Qwen3-30B), and (ii) additional benchmarks (PEToolBench and PrefEval), covering personalized tool use and multi-turn dialogue. The results show consistent improvements across model scales and settings, providing further evidence of generalization.
> > >
> > > We hope these clarifications and additional results help address your concerns. If there are still aspects that you feel remain insufficiently addressed, we would greatly appreciate it if you could point them out more specifically. We would be very happy to further clarify or provide additional evidence.

---

### Official Review · Reviewer_wHVw · 2026-03-13

**Soundness:** 3
**Presentation:** 3
**Significance:** 4
**Originality:** 4
**Overall Recommendation:** 6
**Confidence:** 4

**Summary:**

The paper mentioned that not all tokens in a reply are equally important: some merely serve to convey basic meaning, whilst others truly embody personalised style, preferences or character traits. Current training methods typically treat all tokens equally, which dilutes the significance of these truly important ‘personalised tokens’. The paper first proposes a token-level metric: PIR (Personal Influence Ratio). For the $i$-th output token, compare the difference in log-probabilities between the case ‘with persona’ and the case ‘without persona’. The larger the difference, the more the token relies on user information, and the more it resembles a ‘personalised token’. The author then incorporated this score as a weight into the training process.

**Compliance With Llm Reviewing Policy:**

Affirmed.

**Key Questions For Authors:**

no other questions, check the weaknesses.

**Limitations:**

yes

**Strengths And Weaknesses:**

### Strengths
1. Reframing the personalisation problem as a token-level problem is highly innovative and of great significance to research into the personification of large language models.
1. PerCE requires no additional labelling and does not alter the model architecture; it simply performs an extra ‘de-personalisation’ forward pass during training, before using PIR to compute the weights. From an engineering perspective, it is easier to integrate than many new architectural approaches. The paper also explicitly emphasises that it is orthogonal to and compatible with existing personalised training workflows.
1. The authors recognised that there were no pre-existing labels for ‘token-level personalisation’, so they constructed a synthetic dataset specifically for this purpose, treating the question of whether a token was personalised as a binary classification problem for validation. The results show that PerContrast outperforms both word matching and LLM-as-a-judge in terms of precision, recall and F1 score; Figure 5 also demonstrates a clear monotonic relationship between PIR and personalised tokens.

### Weaknesses
1. The authors have compared PerCE with LossCE and EntCE, which is a more rigorous approach than simply comparing it with CE alone. However, one question remains open: to what extent does PerCE’s advantage stem from ‘true personalisation’, and to what extent from the fact that ‘any reweighting that identifies the key token will be effective’? The paper leans towards the former, but has not yet fully disentangled the two.

---

> ### Author Rebuttal · Authors · 2026-03-31
>
> We appreciate your positive evaluation of our work, particularly your recognition of **its novelty, practical significance, and compatibility with existing personalized training workflows**. We also thank you for raising this insightful question, which we address below.
>
> ---
>
> W1. The authors have compared PerCE with LossCE and EntCE, which is a more rigorous approach than simply comparing it with CE alone. However, one question remains open: to what extent does PerCE’s advantage stem from *‘true personalisation’*, and to what extent from the fact that ‘any reweighting that identifies the key token will be effective’? The paper leans towards the former, but has not yet fully disentangled the two.
>
> This is a deep and important question. To address it, we first need to clarify what *“true personalization”* means.
>
> We view personalization as an **overlay objective**: the goal is for the LLM to adapt to user preferences **on top of** performing various basic tasks. As a result, both training and evaluation inherently involve two entangled objectives—the base task itself and personalization—and these cannot be cleanly decoupled. Therefore, it is not possible to directly train for, or directly measure, *“true personalization”* in isolation.
>
> From the training perspective, our method aims to **shift the mixed objective closer to personalization** by reweighting tokens according to their token-level personalization relevance. In this sense, PerCE makes optimization more biased toward *true personalization*. However, since personalization is always coupled with a base task, we also cannot design a fully isolated test for *true personalization*.
>
> We therefore adopt an **indirect evaluation criterion**. Our intuition is that if a model has truly learned personalization, this ability should transfer across different base tasks and scenarios, rather than being limited to a specific dataset or task format. This is why we evaluate **cross-task** and **cross-scenario** personalization, as shown in Tables 3 and 4 of the paper.
>
> To further support this point, we additionally evaluate a Qwen3-4B checkpoint trained on the LongLaMP topic writing task on **PENGUIN** [1], a personalized safety benchmark. The results are shown below:
>
> |  | Career | Education | Financial | Health | Life | Relationship | Social | ave |
> | --- | --- | --- | --- | --- | --- | --- | --- | --- |
> | CE | 2.35 | 2.15 | 2.28 | 2.15 | 2.30 | 2.07 | 2.13 | 2.21 |
> | LossCE | 3.27 | 3.31 | 3.06 | 3.47 | 3.19 | 3.13 | 3.17 | 3.23 |
> | EntCE | 3.15 | 3.35 | 3.03 | 3.53 | 3.17 | 3.15 | 3.08 | 3.21 |
> | PerCE | 3.75 | 3.82 | 3.72 | 4.03 | 3.71 | 3.89 | 3.72 | 3.81 |
>
> Compared with standard CE and other key-token reweighting losses, PerCE shows clearly stronger **transferable personalization**. We view this as indirect evidence that PerCE’s advantage is not merely due to generic key-token reweighting, but rather comes from pushing optimization closer to personalization itself.
>
> [1] Personalized Safety in LLMs: A Benchmark and A Planning-Based Agent Approach, NeurIPS 2025
>
> ---
> Thanks again, and we hope the response can fully address your concern!

---

> > ### Author Rebuttal · Reviewer_wHVw · 2026-03-31
> >
> > fully resolved. According to the instructions, I should give a higher score. However, I’m afraid that due to system restrictions, I am unable to award a higher score.

---

> > > ### Author Response · Authors · 2026-04-01
> > >
> > > Thank you for your supportive feedback. We are pleased to learn that your concerns have been fully addressed. We sincerely appreciate your recognition and are grateful for your time and thoughtful review of our work.

---

### Decision · Program_Chairs · 2026-04-30

**Decision:**

Accept (regular)

**Comment:**

This paper introduces PerContrast and PerCE, a novel framework that shifts LLM personalization from the sequence level to the token level by identifying and upweighting tokens with high personal influence. Reviewers across the board found the approach innovative, technically sound, and highly practical due to its compatibility with existing training workflows. The authors successfully addressed initial concerns regarding generalization and computational overhead by providing additional experiments on larger models like Qwen3-30B and diverse benchmarks such as PENGUIN and PrefEval.  Consequently, I recommend this submission for acceptance.